



# A multiproxy database of western North American Holocene paleoclimate records

Cody C. Routson[1*], Darrell S. Kaufman[1], Nicholas P. McKay[1], Michael P. Erb[1], Stéphanie H. Arcusa[1], Kendrick J. Brown[2,3], Matthew E. Kirby[4], Jeremiah P. Marsicek[5], R. Scott Anderson[1], Gonzalo Jiménez-Moreno[6], Jessica R. Rodysill[7], Matthew S. Lachniet[8], Sherilyn C. Fritz[9], Joseph R. Bennett[10], Michelle F. Goman[11], Sarah E. Metcalfe[12], Jennifer M. Galloway[13], Gerrit Schoups[14], David B. Wahl[15], Jesse L. Morris[16], Francisca Staines-Urias[17], Andria Dawson[18], Bryan N. Shuman[19], Daniel G. Gavin[20]

[1] Northern Arizona University, School or Earth and Sustainability, Box 4099 Flagstaff, AZ 86011 USA
[2] Canadian Forest Service, Natural Resources Canada, Victoria, BC V8Z 1M5, Canada
[3] University of British Columbia, Okanagan. Department of Earth, Environmental and Geographic Sciences, Canada
[4] Cal-State Fullerton, Department of Geological Sciences, 800 N. State College Blvd., Fullerton, CA 98324, USA
[5] University of Wisconsin-Madison, Department of Geoscience, 1215 W. Dayton St. Madison, WI 53706, USA
[6] Universidad de Granada, Spain, Departamento de Estratigrafía y Paleontología, Avda. Fuentenueva S/N, Granada 18002, Spain
[7] United States Geological Survey, Florence Bascom Geoscience Center, 12201 Sunrise Valley Dr. MS926A, Reston, VA, 20192, USA
[8] University of Nevada Las Vegas, Department of Geoscience, 4505 S. Maryland Parkway, Las Vegas, NV, 89154
[9] University of Nebraska-Lincoln, Department of Earth and Atmospheric Sciences, Lincoln, 68588-0340, USA
[10] Carleton University, Department of Biology, 1125 Col By Drive, Ottawa, ON, K1S 5B6, Canada
[11] Sonoma State University, Department of Geography, Environment, and Planning, 1801 E.Cotati Ave, Rohnert Park, CA, 94928, USA
[12] University of Nottingham, School of Geography, University Park, Nottingham, Nottinghamshire, NG7 2RD, UK
[13] Geological Survey of Canada/Commission gÈologique du Canada, 3303 33rd St. NW, Calgary, AB, T2L 2A7, Canada
[14] Delft University of Technology, Water Resources Management, PO Box 5048, Delft, 2600 GA, Netherlands
[15] United States Geological Survey, GMEG Science Center, 345 Middlefield Rd., Menlo Park, CA, 94025, USA
[16] University of Utah, Department of Geography, 260 Central Campus Dr #4625, Salt Lake City, UT, 84112, USA
[17] Geological Survey of Denmark and GreenlandóGEUS, Department of Marine Geology, Oester Voldgade 10, Copenhagen K, 1350, Denmark
[18] Mount Royal University, Department of General Education, 4825 Mt Royal Gate SW, Calgary, T3E6K6, Canada
[19] University of Wyoming, Geology and Geophysics, 1000 E. University Ave., Laramie, Wyoming, 82071,USA
[20] University of Oregon, Department of Geography, 1251 University of Oregon, Eugene OR, 97403, USA

*correspondence to Cody C. Routson (cody.routson@nau.edu)



**Abstract**

Holocene climate reconstructions are useful for understanding the diverse features and spatial
heterogeneity of past and future climate change. Here we present a database of western North American
Holocene paleoclimate records. The database gathers paleoclimate time series from 209 terrestrial and
marine sites, including 382 individual proxy records. The records span at least 4,000 of the last 12,000
years (median duration = 10,603 years), and have been screened for resolution, chronologic control, and
climate sensitivity. Records were included that reflect temperature, hydroclimate, or circulation features.
The database is shared in the machine readable Linked Paleo Data (LiPD) format and includes
geochronologic data for generating site-level time-uncertain ensembles. This publicly accessible and
curated collection of proxy paleoclimate records will have wide research applications, including, for
example, investigations of the primary features of ocean-atmospheric circulation along the eastern margin
of the North Pacific and the latitudinal response of climate to orbital changes. The database is available
for download at: https://doi.org/10.6084/m9.figshare.12863843.v1 (Routson and McKay, 2020).

### 1.      Introduction

Reconstructing past climate is challenging because it is spatially and temporally complex and because all
paleoclimate records are influenced by factors other than climate. Although rarely done, taking
advantage of the full breadth of paleoclimatic evidence provides the best possibility of discerning signal
from noise. Of all the geologic epochs, the paleoclimate of the Holocene (11.7 kilannum (ka) to present)
has been investigated most extensively. For example, a keyword search on "Holocene" and "climate"
returns approximately 21,000 studies globally on the Web of Science. The volume of this previous work,
and the evolving scientific understanding that it represents, generates organizational challenges related to
data validation, extraction, and application.

Here we present a new database of Holocene paleoclimate records from western North America and the
adjacent eastern Pacific Ocean. The spatial domain (Figure 1) extends from tropical Mexico to Arctic
Alaska. This region was chosen because: 1) it encompasses the large latitudinal range necessary to study
effects of orbital changes, the primary climate forcing during the Holocene; 2) it is affected by the major
modes of modern Pacific climate variability including the Pacific Decadal Oscillation (Mantua et al., 1997),
El Niño Southern Oscillation (ENSO) (Redmond and Koch, 1991), and the Northern Annular Mode
(McAfee and Russell, 2008), among others; 3) it represents a range of climatologies, especially
hydroclimate as influenced by the Pacific westerlies and North American Monsoon (Adams and Comrie,
1997); 4) it features multiple sources of proxy climate information, including marine sediment, caves,
glaciers, and lakes, which are sensitive to changes in wintertime moisture, a key variable for tracking the
primary variability of North Pacific ocean-atmospheric circulation; and 5) it is a region of concern for





future climate change, considering the large population growth and climate-hazards related to, for

example, water scarcity in the southern tier (Garfin, 2013) and changing wildfire hazards throughout (e.g. Marlon et al., 2012; Power et al., 2008).

This database is composed of records from individual site-level studies, and records that were compiled by previous summaries. Many (42%) of the records in this database are also included in version 1 of the global Temperature 12k database (Kaufman et al., 2020a). This database adds another 39

temperature-sensitive records, plus 179 records that reflect hydroclimate and circulation changes. The added data were published in various formats, and often with little metadata to inform the reuse of the data. Together, this geographically distributed collection of proxy climate records integrates marine and terrestrial realms, and forms a network from which to assess the spatial variability of regional climatic change and ocean-atmospheric circulation, and to compare with climate model simulations of past

climate states.

## 2.    Data and Methods

### 2.1 Data collection

Paleoclimate records located in western North America and the adjacent Pacific Ocean (Figure 1) were considered. They were obtained from public archives in PANGEA and NOAA's World Data Service for

Paleoclimatology using the keyword search "Holocene" and record duration searches on NOAA's paleoclimate search engine. The remainder were obtained through either the supplements of publications, or directly from individual data generators and are now being made available in digital form as part of this data product. This database builds on several previously published paleoclimate data compilations overlapping the spatial domain encompassed by this study. These include the global

Holocene temperature reconstruction of Marcott et al. (2013) (n = 4 records in western North America), Arctic Holocene Transitions Database (Sundqvist et al., 2014) (n = 30 records in western North America), a collection compiled to characterize Holocene North American Monsoon variability (Metcalfe et al., 2015) (n = 8 records in common with this database), the Northern Hemisphere dataset used to reconstruct Holocene temperature gradients and mid-latitude hydroclimates (Routson et al., 2019) (n = 55 records in

common with this database), a network of Holocene pollen reconstructions (Marsicek et al., 2018), (n = 71 records in common with this study), two collections of records focused on the last two millennia (Rodysill et al., 2018; Shuman et al., 2017) (n = 18 and n = 16 records in common with this study respectively), and the global Temperature 12k database (Kaufman et al., 2020a) (n = 161 records in common with this database). Two dust deposition records were included from the global dust compilation (Albani et al.,

2015). This database also complements the recently published PAGES global multiproxy database for temperature reconstructions of the Common Era (PAGES 2k Consortium, 2017), and the PAGES global



database for water isotopes over the Common Era (Konecky et al., 2020), which are both structured in the same format as this database. A few of the records were not available from the original data generators and therefore the time series data were digitized from the source publication (as noted in the metadata). These were mainly included to fill geographic gaps in the network of proxy sites.

Other Holocene paleoclimate records were considered but ultimately excluded because they did not satisfy the selection criteria. The majority of excluded records either (1) lacked a clear relation between proxy and climate; (2) were of insufficient duration; (3) possessed large gaps between chronologic control points; or (4) did not meet the sampling resolution criteria. In some instances selection criteria were eased to fill geographic gaps, or for reasons justified by the authors in the '*QC Comments*' metadata. Removing records from the database for subjective reasons, such as removing records with outliers, was avoided.

### 2.2 Relation between proxy and climate

Only records with a demonstrated relation to a climate variable were included, as interpreted by the original authors of the site-level studies, but some records are not calibrated to a climate variable. Calibrated records, for example, are presented in temperature units (°C) and precipitation units (mm). Other records are reported in their native proxy variables (e.g., d$^{18}$O, ‰, or sediment mass accumulation, g/cm$^2$/yr). Some calibrated records rely on statistical procedures to determine the relationship between proxy and instrumental data and to infer palaeoclimate change, assuming that the processes that control the proxy signal remain constant down core (Tingley et al., 2012; Von Storch et al., 2004). Other calibrations rely on transfer functions based on the correlation of contemporary environmental gradients (e.g. Juggins and Birks, 2012), or the modern analogue technique, which uses the similarity between modern and fossil assemblages (e.g. Guiot and de Vernal, 2007). The original species assemblage data (primarily pollen) for these records are not included in this data product, However a link to the Neotoma Paleoecology Database dataset ID is provided where available. The Neotoma Paleoecology database is a community-curated database that is a primary repository for assemblage and other paleoecology data (Williams et al., 2018).

The database also includes proxy records that have not been calibrated to a specific climate variable, but that display a clear relation between the proxy and climate. These "relative" climate indicators are useful because they: 1) attest to the timing and relative magnitude of change, which is sufficient for many statistical reconstruction methods, especially those that do not assume linearity between proxy and climate variables; 2) can be used in proxy system modelling and in some cases (e.g., δ$^{18}$O) can be compared directly to the output of climate models; and 3) provide more complete spatial coverage.

### 2.3 Record duration and resolution



The database aims to document paleoclimate variability that ranges in time-scale from multi-millennial
trends to centennial excursions. However, not all records encompass the entire Holocene epoch. To be
included, records must span a duration of ca. 4,000 years anytime between 0 and 12 ka. To focus on
records that can resolve sub-millennial patterns, the database includes those with sample resolution finer
than 400 years (i.e., the median spacing between consecutive samples in the time series is less than 400
years over the past 12,000 years or over the full record length, if shorter).

**2.4 Chronologic control**

Age control is a fundamental variable underlying proxy records. The database includes the chronologic
data necessary for reproducing original age-depth models for records from sediment and speleothem
archive types. Chronologic data include depth, uncalibrated radiometric or other dates, analytical errors,
and associated corrections where applicable. Other metadata, including material type analyzed and
sample identifiers, were included when available. Time series with a maximum of 3,000 years between
dates within the 0-12 ka interval or with five or more relatively evenly distributed Holocene dates were
included in the database. Overall, the age-control screening retained a high proportion of available
records, while recognizing that such coarse age control often precludes the ability to address questions
that require fine temporal-scale accuracy (Blaauw et al., 2018).

**2.5 Metadata**

The database includes a large variety of metadata (Supplementary Table 1) to facilitate analyses and
re-use. The metadata included in this database are largely consistent with those developed and used in
the Temperature 12k database (Kaufman et al., 2020), with some refinement for hydroclimate related
records. Predominant metadata are subdivided into the following categories:

(1)   Geographic information includes '*Site Name*', '*Latitude*', '*Longitude*', and '*Elevation*'. Geodetic data
          are relative to the WGS84 ellipsoid and in units of decimal degrees. '*Country Ocean*' is generated
          based on NASA GCMD convention.

       (2)   Bibliographic information includes the DOI when available. The original study is typically
          referenced in '*Publication 1*' . '*Publication 2*' generally corresponds to subsequent publications
170       contributing to record development or reuse.

       (3)   Original data source '*Original Data Citation*' is the persistent identifier (URL or DOI) that connects
          to the publicly accessible repository (e.g. PANGAEA and WDS-NOAA Paleoclimatology when
          available). Fields with the entry '*wNAm*' correspond to records transferred to a public repository



for the first time by this study. '*Neotoma ID*' includes the Neotoma dataset ID when available for the original assemblage data.

(4) Metadata describing the proxy record include '*Archive Type*', '*Proxy General*', '*Proxy Type*', '*Proxy Detail*', '*Calibration Method*', and '*Paleo Data Notes*'. '*Archive Type*' corresponds to the physical archive (e.g. lake sediment, marine sediment, peat, speleothem). '*Proxy General*' simplifies plotting figures by grouping similar proxies from '*Proxy Type*'. For example, '*Proxy General*' = '*other biomarkers*' includes '*Proxy Type*' TEX86 and GDGT, but not alkenones, which are treated separately. '*Proxy General*' = 'biophysical' includes biogenic silica, tree-ring width, total organic content, chlorophyll and macrofossils. '*Proxy General*' = '*other microfossil*' includes coccolith, diatom, dinocyst, and foraminifera. Pollen and chironomid records are treated separately. '*Proxy Detail*' corresponds to specific species or material types. '*Calibration Method*' is the statistical method used for proxy calibration. '*Paleo Data Notes*' includes information from the original study to help users understand the proxy record.

(5) Climate interpretation. Records included in this collection have been interpreted in a peer-reviewed publication as reflecting past climate variability. Primary '*Climate Variables*' include '*T*' (temperature), '*P*' (precipitation), and '*P-E*' (precipitation minus evaporation). Other climate indicators include '*MODE*' (climate modes such as ENSO), '*upwelling*' (coastal upwelling), '*DUST*' (dust deposition), '*ICE*' (sea ice extent), and '*ELA*' (glacier equilibrium line altitude). The '*Interpretation Direction*' is the sign relation ('positive' or 'negative') between the proxy value and the '*Climate Variable*'. Proxy records originally reported as E-P were cataloged as '*Climate Variabile*' = P-E, and the field '*Interpretation Direction*' was inverted from the original interpretation. '*Variable Name*' corresponds to the specific variable type (e.g. '*temperature*', or '*d18O*'). '*Units*' correspond to the measurement unit specified in '*Variable Name*' (e.g. '*degC*' or '*permil*'). '*Climate Variable Detail*' refines the '*Climate Variable*' field. Temperature records follow the structure of the variable sensed (e.g. '*air*') at a specific level (e.g. '*surface*'). Examples include '*air@surface*', '*air@condensation*', and '*sea@surface*'. Hydroclimate and some other record types do not always conform as well to this format. '*Climate Variable Detail*' for these records specifies the variable sensed (e.g. '*lake level*', '*runoff*', '*river flow*', '*amount*'), at a specific level (e.g. '*surface*'). Examples include '*lakeLevel@surface*' and '*runoff@surface*'. If the variable sensed is the same as the '*Climate Variable*' (e.g. 'precipitation'), the field is left blank. In these cases only the level is specified (e.g. '@surface'). In cases where the level was ambiguous, not specified, or not applicable (e.g. '*soil moisture*', '*lake salinity*', '*El Nino*'), only the variable sensed was specified.

(6) Seasonality information has been separated into two fields '*Seasonality*' and '*Seasonality General*'. '*Seasonality*' includes the most specific seasonal information available including specific months





in number format (July = '7'), or reconstructed seasons (e.g. '*Warmest Month*', '*Summer*', '*Growing Season*', '*Winter*', '*Annual*'). '*Season General*' distills season details into queryable seasons ('*Annual*', '*Summer Only*', '*Summer+*', '*Winter Only*', '*Winter+*'). Categories '*Summer+*' and '*Winter+*' indicate another season (or annual) has also been reconstructed from the same site.

(7) Metadata describing the underlying time-series data include the youngest and oldest sample ages ('*Min Year*' and '*Max Year*'), *the* median sample resolution ('*Resolution*') over the past 12,000 years, *and* the frequency of age control points ('*Ages Per kyr*'), which includes radiocarbon and U-series ages.

(8) Quality control metadata includes ('*QC Certification*') and ('*QC Comments*'). '*QC Certification*' includes initials of the co-author of this data descriptor who was responsible for reviewing the screening criteria for records included in the data product. '*QC Comments*' were written by the QC'er to improve reusability of the data.

(9) Data access and visualization includes a website link for viewing and downloading the data in .csv or LiPD format ('*Link to LiPDverse*').

**2.6 Database structure and format: Linked Paleo Data (LiPD)**

The site-level data and metadata are formatted in the LiPD structure. The LiPD framework comprises JSON formatted files that are machine-readable with MatLab, Python, and R packages that enable rapid querying and data extraction (McKay and Emile-Geay, 2016). LiPD encodes the database into a structured hierarchy that allows explicit descriptions at any level and aspect of the database. Code packages for evaluating the database can be accessed on GitHub (https://github.com/nickmckay/LiPD-utilities).

**2.7 Data visualization**

A one-page dashboard for each record is included as a supplement to this article. The dashboards include the primary information associated with each record including the location, the time-series plot, bibliographic reference, and proxy data information (Supplemental Dashboards). Each record is also linked to a webpage ('*Link to LiPDverse*') where the data can be visualized and downloaded in LiPD or text versions. A globally distributed collection of paleoclimate LiPD files is housed at LiPDverse.org. This western North American Holocene paleoclimate database is a subset of the records that can be found by choosing "wNAm" in the LiPDverse browser. The full collection can also be accessed at http://lipdverse.org/wNAm0_15_0/.

**3.    Summary of Database Contents**

### 3.1 Proxy records and climate variables

The western North American Holocene paleoclimate database includes proxy climate records from 120
different sites. Many "sites" (locations) are represented by more than one proxy "record" (time series).
Multiple records from one site often represent different climate variables or reconstruction methods.
Pollen assemblages, for example, are often translated into both temperature and moisture variables,
sometimes for different seasons. The list of sites is shown by row in Table 1, whereas Supplementary
Table 1 contains a row for each record. In total, this database comprises 209 sites and 382 records.

The records are derived from 9 archive types, and are based on 8 proxy categories (Supplementary Table
1). The database includes 255 records from lake sediments, 63 records from marine sediment, and 64
other terrestrial.

The western North America database includes 61 'new' records that are being transferred to a publicly
accessible data repository for the first time with this data product. These include 23 pollen ratio time
series reflecting changes in the position of forest boundaries and long-term temperature change. These
ratios were computed by the original data generators following methods and rationale described in
Jiménez-Moreno et al. (2019) and Johnson et al. (2013). The database also includes 20 precipitation
records, which were generated by Marsicek et al. (2018), but not released with that publication. Finally,
we have included 18 hydroclimate records based on subsets of packrat midden sites from Harbert et al.
(2018), following the same methods applied for temperature reconstructions in Kaufman et al. (2020a).
These records are noted in the '*QC comments*' column of Supplementary Table 1.

The database contains 200 temperature sensitive records, 152 hydroclimate sensitive records
(precipitation, P-E, flood frequency, streamflow), and 27 other records including upwelling, dust, climate
mode, and sea-ice extent. Marine records are primarily sea surface temperatures, but there are several
marine records of other variables including sea ice extent, upwelling strength, and flood frequency. Many
(224) of the proxy records are interpreted by the original authors to represent mean annual values of
specific climate variables. Others represent individual seasons, primarily some aspect of summer.

### 3.2 Geographic coverage

The geographic distribution of records within western North America is far from uniform (Figure 1). The
density of all sites is comparatively high in Alaska and the conterminous western United States. In
contrast, Mexico is represented by few study sites, mainly because many studies failed to meet the
inclusion criteria. Hydroclimate records have the most uniform coverage, albeit with a spatial gap in
Mexico. The spatial distribution of temperature records has gaps in Canada, the mid-western United
States, Texas, and continental Mexico.



### 3.3 Record length and temporal resolution


Median record duration is 10,603 years, not counting the duration of records beyond 12,000 years. Most of the records (94%) extend back at least 6,000 years, thereby including the frequently modeled 6 ka paleoclimate time slice. The median sample resolution of individual records in the database is 128 years (Figure 2).

### 3.4 Geochronology


Original geochronologic data for each record are included in the database. The database includes 2356 individual age control points ($^{14}$C, $^{210}$Pb, tephras, etc.). Tree-ring age control points (two studies) were excluded from this number. These primary age control can be used to recalculate the age models for all of the $^{14}$C-based sedimentary sequences and U-series-based speleothems using a systematic approach to

addressing age uncertainty.

### 3.5 Uncertainties

A variety of approaches have been used to characterize uncertainties in paleoclimate variables and there is no standard procedure for either calculating or reporting uncertainties (Sweeney et al., 2018). Generally, calibration and other uncertainties are large relative to the small amplitude of most Holocene

climate change, but these uncertainties are less important when investigating the relative magnitude of climate changes rather than the absolute value of a climate variable. Uncertainty arising from differences among records can be explored using a bootstrapped sampling with replacement approach (e.g. Boose et al., 2003; Routson et al., 2019), however these ranges reflect a combination of record-level uncertainty and regional climate heterogeneity. In this database we are following other syntheses (Kaufman et al., 2020b;

Marcott et al., 2013; Routson et al., 2019) by applying a single uncertainty estimate for each proxy type (Supplementary Table 1). Proxy specific uncertainties for temperature records follow Kaufman et al. (2020b), as did our approach for calculating uncertainty estimates for the hydroclimate records. For the calibrated hydroclimate records (primarily pollen based), we have calculated average RMSE values from the following references within or adjacent to the study region (Brown et al 2006; 2015; 2019; Herbart et

al., 2018; Marsicek et al., 2013). For the 166 uncalibrated records we have estimated the error as ±1 SD of the Holocene values.

### 3.6 Summarizing major trends

Recognizing major climatological differences across the study domain (spanning from tropical Mexico to Arctic Alaska), we have summarized some dominant patterns in the database including climate variables

(temperature and hydroclimate), proxy group, and season. Dominant temperature and hydroclimate patterns by proxy group as specified in Supplementary Table 1, '*Proxy General*' were evalutaed (Figure 3).





Only proxy groups with more than 10 records were considered. The records were screened by season to include one record per site ('*Season General*' = '*annual*', OR '*summerOnly*', OR '*winterOny*'). Records were then binned to 500-year resolution by averaging data points within respective intervals, normalized to a
mean of zero and 1 SD variance (z-scores), and composited using the median to minimize the influence of outliers. Temperature proxies include chironomids (n = 15), biophysical (n = 15), pollen (n = 66), and isotopes (n = 14). Chironomids show peak warmth in the early Holocene (ca. 10 ka), followed by a Holocene cooling trend. Biophysical records have more variability, with peak warming ca. 7 ka. Pollen records show relatively low Holocene variability, with peak warming at ca. 6 ka. Isotopes have the
highest Holocene variability and the lowest sample depth, and show two intervals of warming (ca. 9 ka and 4 ka). Hydroclimate proxies include other microfossil (n = 11), biophysical (n = 46), pollen (n = 55), and isotopes (n = 35). Other microfossils show variable Holocene conditions, with the wettest period in the early Holocene. This interval however, has very low sample depth. Biophysical records show only small Holocene hydroclimate changes. Pollen records show a strong Holocene wetting trend. Whereas
isotope records show variable conditions.

Temperature and hydroclimate trends were compared by summer, winter, and annual seasons (Figure 4). The records were binned to 500-year resolution by averaging data points within respective intervals and normalized to a mean of zero and 1 SD variance (z-scores). Records were then averaged into equal-area (127,525 km$^2$) grids following Routson et al., (2019). The grids were then combined into a single
composite using the median. The most recent 500-year bin was then subtracted, registering the present end to zero. This was done to help compare the seasonal Holocene evolutions. In the early to middle Holocene (ca 12 ka to 6 ka), summertime and annual temperatures warmed faster than wintertime temperatures, consistent with Northern Hemisphere seasonal insolation forcing (Berger and Loutre 1991). Temperatures in all seasons show a cooling pattern from ca. 6 ka to the present. Hydroclimate composites
show a Holocene-length wetting trend in all seasons, with the largest trend in wintertime.

### 4.  Use and Limitations

The machine-readable database includes multiple parameters for searching and screening records. The data compilation will form the foundation of new analyses of Holocene climate variability in western North America and will help identify future research priorities. The 382 records in this database will
enable studies of Holocene climate on centennial to multi-millennial time scales. At finer time scales, the number of records with sufficient resolution and geochronological control is more limited. For example, 168 records have a median sampling resolution of better than 100 years, and only 25 sites have resolution finer than 10 years. The accuracy and precision of age control can also limit inferences involving correlations and spectral properties of the time series. The availability of the raw chronology data for each



record in this database allows users to quantify and incorporate aspects of chronologic uncertainty into their analyses.

This database represents a concerted effort to generate a comprehensive data product, but is an ongoing effort, with newly published records continuing to be added. Some published records that meet the criteria might have been inadvertently overlooked. Readers who know of missing datasets, or who find

errors in this version are asked to contact one the authors so that future versions of the database will be more complete and accurate. Rather than issuing errata to this publication, errors and additions will be included in subsequent versions of the database.

**Data and code availability:** The database is available for download at:
https://doi.org/10.6084/m9.figshare.12863843.v1 (Routson and McKay, 2020), with serializations for

MatLab and R. Supplementary Table 1 lists the essential metadata. Data can also be viewed and accessed at http://lipdverse.org/wNAm0_15_0. Code, including basic functions for analyzing LiPD files in three programming languages, is available on GitHub (https://github.com/nickmckay/LiPD-utilities).

**Acknowledgements:** Funding for this research was provided by the U.S. National Science Foundation (AGS-1602105 and AGS-1903548). We also thank the U.S. Geological Survey Powell Center for Analysis
and Synthesis who hosted a meeting that led to this synthesis effort. Any use of trade, firm, or product names is for descriptive purposes only and does not imply endorsement by the U.S. Government. We thank the original data generators who made their data available for reuse and we acknowledge the data repositories for safeguarding these assets.

**Author Contributions:**

CCR led the project, data collection, and data formatting. CCR, DSK, MPE, NPM, MEK, JPM, FSU, MSL, SHA, JRB, MFG, SEM, KJB, JMG, SCF, GS, JRR, JLM, DBW, RSA, BNS, and GJM contributed and certified data. CCR and MPE analyzed the database and produced the figures. NPM built the data infrastructure and performed data processing. CCR, DSK, and SHA did quality control, term standardization, and database cleaning. CCR and DSK wrote the manuscript with contributions from the other authors.

**A**



**Figure 1.** Spatiotemporal distribution of the western North American Holocene paleoclimate database. A)





The database includes 382 proxy records from a variety of archive and proxy types. Records include those in calibrated climate units (e.g. °C) and records in their native proxy units (e.g. δ¹⁸O). B) Distribution of records sensitive to hydroclimate including precipitation, flood frequency, and P-E (n = 152). C) Spatial distribution of the subset of records sensitive to temperature (n = 200), and D) the spatial distribution of other records including upwelling, sea ice, glacier extent, dust, circulation, and climate modes (n = 27). E) Temporal availability of the records in the database by proxy type (Supplemental Table 1, '*Proxy General*') over the last 12 ka.

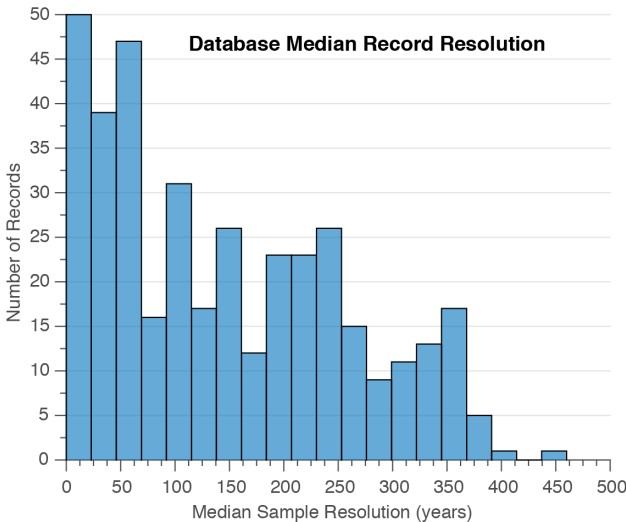

**Figure 2:** Median sample resolution for all records in the database (20-year intervals).






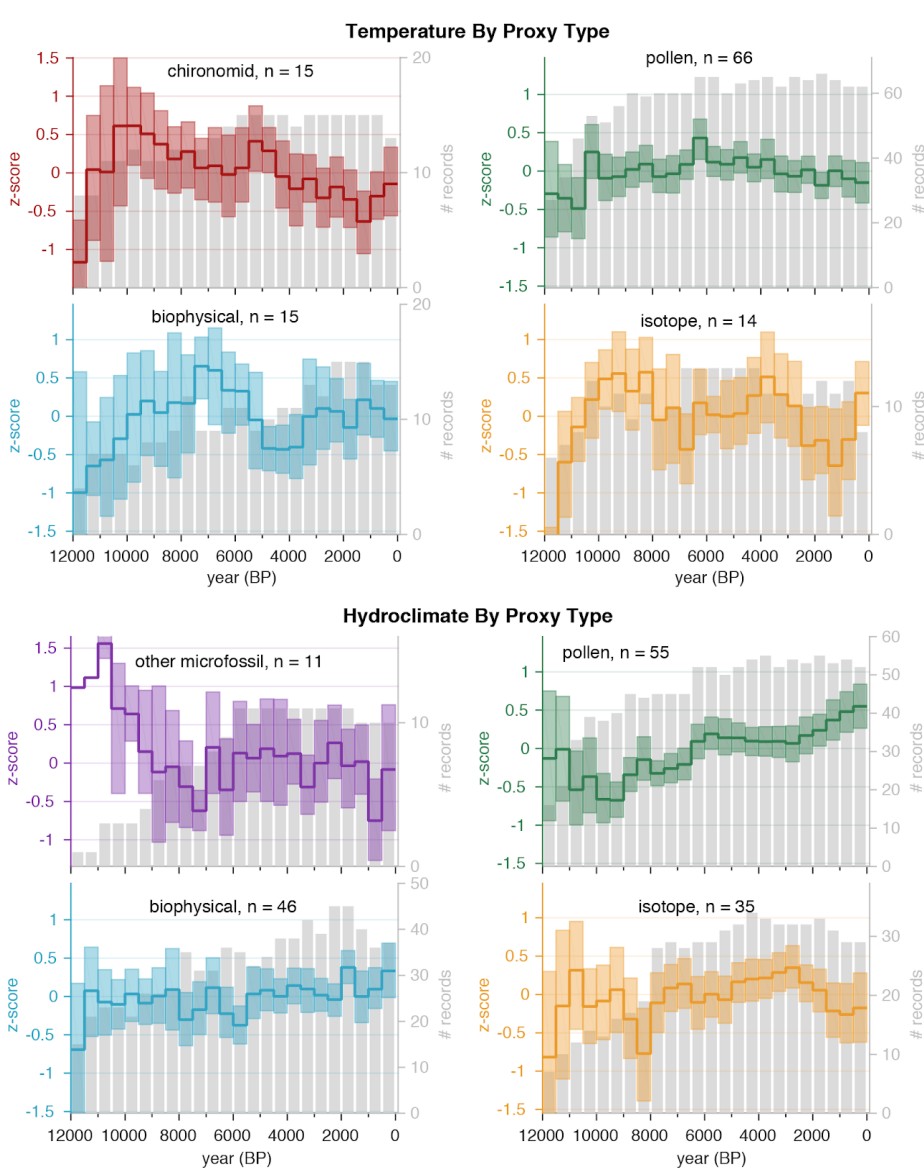

**Figure 3:** Temperature(top) and hydroclimate (bottom) composites by dominant proxy types

(Supplementary Table 1, '*Proxy General*'). Only proxy types with n > 10 are shown. The composites are

produced from normalized (standard deviation units) records to include both calibrated and uncalibrated time series. Records have been filtered by seasonality ('*Season General*' = '*annual*', '*summerOnly*', and '*winterOnly*'), to include one record per site. Shading shows the 95% bootstrapped confidence interval on the estimate on the mean over 1000 sampling with replacement iterations. Gray bars show the number of
records contributing to each 500-year bin.

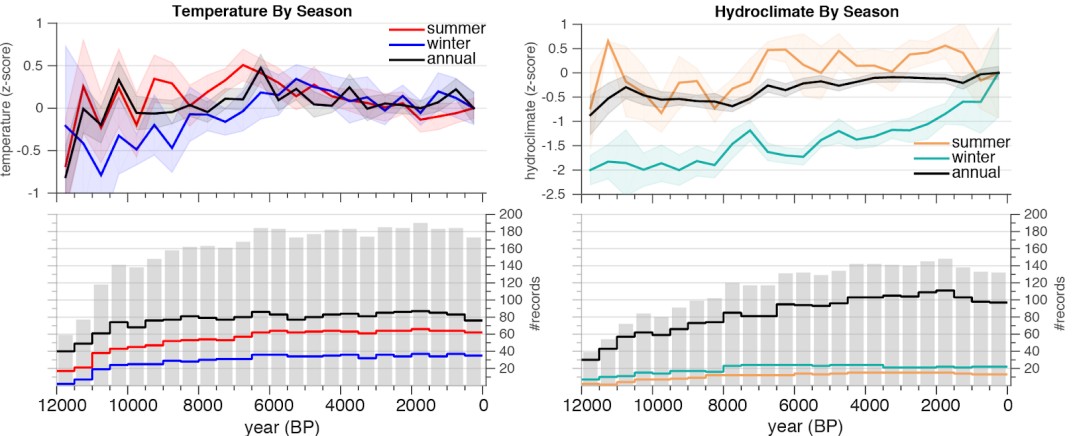

**Figure 4:** Comparison of seasonal temperature (left) and hydroclimate (right) composites. The composites are produced from binned (500-year bins) and normalized (standard deviation units) records averaged on an equal area grid. The most recent bin has been registered to zero to help compare the Holocene trends
with respect to preindustrial conditions. Both calibrated and uncalibrated time series are included. Shading shows the 1 standard deviation bootstrapped confidence interval on the estimate on the mean over 1000 (sampling with replacement) iterations. Gray bars (bottom) show the total number of records (all seasons) in each 500-year bin, whereas the time series (bottom) show the number or records contributing to each composite by color.

**Table 1:** Proxy records included in the database, listed alphabetically. See Supplementary Table 1 for expanded metadata and links to the proxy time series and chronology data.

| Site Name | Lat (°N) | Long (°W) | Archive Type | Proxy[a] | Original Data Citation | Reference |
|---|---|---|---|---|---|---|
| 3M Pond | 49.98 | -121.22 | LakeSediment | chironomid | www.ncdc.noaa.gov/paleo/study/27330 | Pellatt et al., 2000 |
| 893A | 34.29 | -120.04 | MarineSediment | d18O | www.ncdc.noaa.gov/paleo/study/27330 | Kennett et al., 2007 |
| Abalone | 33.96 | -119.98 | LakeSediment | pollen | www.ncdc.noaa.gov/paleo/study/27330 | Cole and Liu, 1994 |
| Alfonso Basin | 24.65 | -110.60 | MarineSediment | coccolith | wNAm | Staines-Urías et al., 2015 |
| Andy | 64.65 | -128.08 | LakeSediment | pollen | www.ncdc.noaa.gov/paleo/study/15444 | Szeicz et al., 1995 |





| | | | | | | |
|---|---|---|---|---|---|---|
| Banks Island -12 | 72.37 | -119.83 | LakeSediment | pollen | www.ncdc.noaa.gov/paleo/study/27330 | Gajewski et al., 2000 |
| Banks Island -15 | 73.53 | -120.22 | LakeSediment | pollen | www.ncdc.noaa.gov/paleo/study/27330 | Gajewski et al., 2000 |
| Battle Ground | 45.80 | -122.49 | LakeSediment | pollen | www.ncdc.noaa.gov/paleo/study/27330 | Barnosky, 1985b |
| Beaver Lake | 42.46 | -100.67 | LakeSediment | diatom | www.ncdc.noaa.gov/paleo/study/23075 | Schmieder et al., 2011 |
| Beef Pasture | 37.47 | -108.16 | LakeSediment | pollen | www.ncdc.noaa.gov/paleo/study/27330 | Petersen, 1985 |
| Begbie Lake | 48.59 | -123.68 | LakeSediment | pollen | wNAm | Brown et al., 2019 |
| Bells Lake | 65.02 | -127.48 | LakeSediment | pollen | 10.21233/N35G6P | Szeicz et al., 1995 |
| Big Lake | 51.67 | -121.45 | LakeSediment | diatom | www.ncdc.noaa.gov/paleo/study/23089 | Cumming et al., 2002 |
| Bison Lake | 39.76 | -107.35 | LakeSediment | d18O | www.ncdc.noaa.gov/paleo/study/10749 | Anderson L., 2011 |
| Blue Lake | 37.24 | -106.63 | LakeSediment | XRF | www.ncdc.noaa.gov/paleo/study/27078 | Routson et al., 2019 |
| Boomerang Lake | 49.18 | -124.16 | LakeSediment | pollen | wNAm | Brown et al., 2006 |
| Boone | 55.58 | -119.43 | LakeSediment | pollen | www.ncdc.noaa.gov/paleo/study/27330 | White and Mathewes, 1986 |
| Candelabra Lake | 61.68 | -130.65 | LakeSediment | pollen | www.ncdc.noaa.gov/paleo/study/15444 | Cwynar and Spear, 2007 |
| Carleton Lake | 64.26 | -110.10 | LakeSediment | chironomid | www.ncdc.noaa.gov/paleo/study/16296 | Upiter et al., 2014 |
| Carp | 45.92 | -120.88 | LakeSediment | pollen | www.ncdc.noaa.gov/paleo/study/27330 | Barnosky, 1985a |
| Cascade Fen | 37.65 | -107.81 | LakeSediment | pollen | www.ncdc.noaa.gov/paleo/study/27330 | Maher, 1963 |
| Castor Lake | 48.54 | -119.56 | LakeSediment | reflectance | www.ncdc.noaa.gov/paleo/study/10310 | Nelson et al., 2011 |
| Castor Lake | 48.54 | -119.56 | LakeSediment | d18O | www.ncdc.noaa.gov/paleo/study/10310 | Nelson et al., 2011 |
| Chichancanab Lake | 19.83 | -88.75 | LakeSediment | CaCO3 | www.ncdc.noaa.gov/paleo/study/5483 | Hodell et al., 1995 |
| Chichancanab Lake | 19.83 | -88.75 | LakeSediment | S | www.ncdc.noaa.gov/paleo/study/5483 | Hodell et al., 1995 |
| Chichancanab Lake | 19.83 | -88.75 | LakeSediment | d18O | www.ncdc.noaa.gov/paleo/study/5483 | Hodell et al., 1995 |
| Chihuahuenos Bog | 36.05 | -106.51 | Peat | pollen | wNAm | Anderson RS et al., 2008a |
| Chitina loess section | 61.54 | -144.38 | Loess | particle size | www.ncdc.noaa.gov/paleo/study/20529 | Muhs et al., 2013 |
| Cleland Lake | 50.83 | -116.39 | LakeSediment | d18O | www.ncdc.noaa.gov/paleo/study/21250 | Steinman et al., 2016 |
| Cleland Lake | 50.83 | -116.39 | LakeSediment | d13C | www.ncdc.noaa.gov/paleo/study/21250 | Steinman et al., 2016 |
| Copley | 38.87 | -107.08 | LakeSediment | pollen | www.ncdc.noaa.gov/paleo/study/27330 | Fall, 1997 |
| Corser Bog | 60.53 | -145.45 | Peat | GDGT | www.ncdc.noaa.gov/paleo/study/15444 | Nichols et al., 2014 |
| Corser Bog | 60.53 | -145.45 | Peat | dD | www.ncdc.noaa.gov/paleo/study/15444 | Nichols et al., 2014 |
| Cottonwood Pass Pond | 38.83 | -106.41 | LakeSediment | pollen | www.ncdc.noaa.gov/paleo/study/27330 | Fall, 1997 |
| Crater Lake | 37.67 | -106.69 | LakeSediment | particle size | wNAm | Arcusa et al., 2020 |
| Crevice Lake | 45.00 | -110.58 | LakeSediment | d18O | wNAm | Whitlock et al., 2012 |
| Crevice Lake | 45.00 | -110.58 | LakeSediment | CaCO3 | wNAm | Whitlock et al., 2012 |



| | | | | | | |
|---|---|---|---|---|---|---|
| CuevaDiablo | 18.18 | -99.92 | Speleothem | d18O | www.ncdc.noaa.gov/paleo/study/10670 | Bernal et al., 2011 |
| Cumbres Bog | 37.02 | -106.45 | LakeSediment | pollen | wNAm | Johnson et al., 2013 |
| Dempster Hwy Peatland | 65.21 | -138.32 | Ice-other | d18O | www.ncdc.noaa.gov/paleo/study/27330 | Porter et al., 2019 |
| DJ6-93SF-6 | 37.63 | -122.37 | MarineSediment | Mg/Ca | wNAm | McGann, 2008 |
| DSDP site 480 | 27.90 | -111.65 | MarineSediment | diatom | www.ncdc.noaa.gov/paleo/study/5855 | Barron et al., 2004 |
| DSDP site 480 | 27.90 | -111.65 | MarineSediment | CaCO3 | www.ncdc.noaa.gov/paleo/study/5855 | Barron et al., 2004 |
| DSDP site 480 | 27.90 | -111.65 | MarineSediment | BSi | www.ncdc.noaa.gov/paleo/study/5855 | Barron et al., 2004 |
| Dune | 64.42 | -149.90 | LakeSediment | d13C | www.ncdc.noaa.gov/paleo/study/13076 | Finney et al., 2012 |
| Eldora Fen | 39.94 | -105.58 | LakeSediment | pollen | www.ncdc.noaa.gov/paleo/study/27330 | noPubOnRecord |
| Eleanor Lake | 47.68 | -124.02 | LakeSediment | BSi | wNAm | Gavin et al., 2011 |
| Emerald Lake | 39.15 | -106.41 | LakeSediment | stratigraphy | www.ncdc.noaa.gov/paleo/study/23079 | Shuman et al., 2014 |
| Emerald Lake | 39.15 | -106.41 | LakeSediment | pollen | wNAm | Jiménez-Moreno, et al., 2019 |
| EN32_PC6 | 26.95 | -91.35 | MarineSediment | Mg/Ca | www.ncdc.noaa.gov/paleo/study/27330 | Flower et al., 2004 |
| EN32_PC6 | 26.95 | -91.35 | MarineSediment | d18O | www.ncdc.noaa.gov/paleo/study/27330 | Flower et al., 2004 |
| Enos Lake | 49.28 | -124.15 | LakeSediment | pollen | wNAm | Brown et al., 2006 |
| EW0408_66JC | 57.87 | -137.10 | MarineSediment | alkenone | www.ncdc.noaa.gov/paleo/study/22400 | Praetorius et al., 2015 |
| EW0408_66JC | 57.87 | -137.10 | MarineSediment | d18O | www.ncdc.noaa.gov/paleo/study/22400 | Praetorius et al., 2015 |
| EW0408_85JC | 59.56 | -144.15 | MarineSediment | alkenone | www.ncdc.noaa.gov/paleo/study/21950 | Praetorius et al., 2015 |
| EW0408_85JC | 59.56 | -144.15 | MarineSediment | d18O | www.ncdc.noaa.gov/paleo/study/21950 | Praetorius et al., 2015 |
| EW0408-87JC | 58.77 | -144.50 | MarineSediment | alkenone | wNAm | Praetorius et al., 2020 |
| Farewell Lake | 62.55 | -153.63 | LakeSediment | Mg/Ca | www.ncdc.noaa.gov/paleo/study/15444 | Hu et al., 1998 |
| Felker Lake | 51.95 | -122.00 | LakeSediment | diatom | wNAm | Galloway et al., 2011 |
| Ferndale | 34.41 | -95.81 | LakeSediment | pollen | www.ncdc.noaa.gov/paleo/study/27330 | Albert and Wyckoff, 1981 |
| Foy Lake | 48.20 | -114.40 | LakeSediment | diatom | www.ncdc.noaa.gov/paleo/study/6188 | Stone and Fritz, 2006 |
| Frozen Lake | 49.60 | -121.47 | LakeSediment | chironomid | www.ncdc.noaa.gov/paleo/study/27330 | Rosenberg et al., 2004 |
| GGC19 | 72.16 | -155.51 | MarineSediment | dinocyst | www.ncdc.noaa.gov/paleo/study/15444 | Farmer et al., 2011 |
| GGC55_JPC56 | 27.47 | -112.10 | MarineSediment | diatom | www.ncdc.noaa.gov/paleo/study/5915 | Barron et al., 2005 |
| Great Basin | 38.00 | -116.50 | Wood | TRW | www.ncdc.noaa.gov/paleo/study/17056 | Salzer et al., 2014 |
| Greyling Lake | 61.38 | -145.74 | LakeSediment | TOC | www.ncdc.noaa.gov/paleo/study/15444 | McKay and Kaufman, 2009 |
| Grutas del Ray Marcos | 15.43 | -90.28 | Speleothem | d18O | www.ncdc.noaa.gov/paleo/study/28351 | Winter et al., 2020 |
| Guaymas Basin | 27.48 | -112.07 | MarineSediment | dD | www.ncdc.noaa.gov/paleo/study/24890 | Bhattacharya et al., 2018 |





| | | | | | | |
|---|---|---|---|---|---|---|
| Guaymas Basin | 27.48 | -112.07 | MarineSediment | dD | www.ncdc.noaa.gov/paleo/study/24890 | Bhattacharya et al., 2018 |
| Gulf of Mexico | 27.18 | -91.42 | MarineSediment | foraminifera | wNAm | Poore et al., 2005 |
| Hail Lake | 60.03 | -129.02 | LakeSediment | pollen | www.ncdc.noaa.gov/paleo/study/15444 | Cwynar and Spear, 2007 |
| Hallet Lake | 61.49 | -146.24 | LakeSediment | TOC | www.ncdc.noaa.gov/paleo/study/15444 | McKay and Kaufman, 2009 |
| Hallet Lake | 61.49 | -146.24 | LakeSediment | BSi | www.ncdc.noaa.gov/paleo/study/15444 | McKay and Kaufman, 2009 |
| Hanging Lake | 68.38 | -138.38 | LakeSediment | pollen | www.ncdc.noaa.gov/paleo/study/27330 | Cwynar, 1982 |
| Harding Lake | 64.42 | -146.85 | LakeSediment | TOC | www.ncdc.noaa.gov/paleo/study/15655 | Finkenbinder et al., 2014 |
| Harding Lake | 64.42 | -146.85 | LakeSediment | MS | www.ncdc.noaa.gov/paleo/study/15655 | Finkenbinder et al., 2014 |
| Heal Lake | 48.54 | -123.46 | LakeSediment | pollen | wNAm | Brown et al., 2006 |
| Hermit Lake | 38.09 | -105.63 | LakeSediment | pollen | wNAm | Anderson RS et al., 2019 |
| Hidden Lake CA | 38.26 | -119.54 | LakeSediment | chironomid | www.ncdc.noaa.gov/paleo/study/27330 | Potito et al., 2006 |
| Hidden Lake CO | 40.51 | -106.61 | LakeSediment | stratigraphy | www.ncdc.noaa.gov/paleo/study/23077 | Shuman et al., 2009 |
| HLY0501 | 72.69 | -157.52 | MarineSediment | dinocyst | www.ncdc.noaa.gov/paleo/study/15444 | de Vernal et al., 2013 |
| Honeymoon | 64.63 | -138.40 | LakeSediment | pollen | 10.21233/N33Q7V | Cwynar and Spear, 1991 |
| Hudson-AK | 61.90 | -145.67 | LakeSediment | chironomid | www.ncdc.noaa.gov/paleo/study/15444 | Clegg et al., 2011 |
| Hunters Lake | 37.61 | -106.84 | LakeSediment | pollen | wNAm | Anderson RS et al., 2008b |
| Jellybean Lake | 60.35 | -134.80 | LakeSediment | d18O | www.ncdc.noaa.gov/paleo/study/5445 | Anderson L. et al., 2005 |
| Jenny Lake | 43.75 | -110.73 | LakeSediment | TIC | www.ncdc.noaa.gov/paleo/study/20128 | Larsen et al., 2016 |
| Jones Lake | 47.05 | -113.14 | LakeSediment | d18O | www.ncdc.noaa.gov/paleo/study/23076 | Shapley et al., 2009 |
| Keele | 64.17 | -127.62 | LakeSediment | pollen | www.ncdc.noaa.gov/paleo/study/27330 | Szeicz et al., 1995 |
| Keystone Iron Bog | 38.87 | -107.03 | LakeSediment | pollen | www.ncdc.noaa.gov/paleo/study/27330 | Fall, 1985 |
| Kirman Lake | 38.34 | -119.50 | LakeSediment | diatom | dataverse.harvard.edu/dataverse/UCLAGMacDonald | MacDonald et al., 2016 |
| Kite Lake | 39.33 | -106.13 | LakeSediment | pollen | wNAm | Jiménez-Moreno and Anderson, 2013 |
| KNR159_JPC26 | 26.37 | -92.03 | MarineSediment | Mg/Ca | www.ncdc.noaa.gov/paleo/study/27330 | Antonarakou et al., 2015 |
| KNR159_JPC26 | 26.37 | -92.03 | MarineSediment | d18O | www.ncdc.noaa.gov/paleo/study/27330 | Antonarakou et al., 2015 |
| Koksilah River | 48.76 | -123.68 | LakeSediment | pollen | wNAm | Brown and Schoups, 2015 |
| Kurupa Lake | 68.35 | -154.61 | LakeSediment | chlorophyll | www.ncdc.noaa.gov/paleo/study/18995 | Boldt et al., 2015 |
| Kusawa | 60.28 | -136.18 | LakeSediment | BSi | www.ncdc.noaa.gov/paleo/study/15444 | Chakraborty et al., 2010 |
| Lac Meleze | 65.22 | -126.12 | LakeSediment | pollen | www.ncdc.noaa.gov/paleo/study/27330 | MacDonald, 1987 |
| Lago Minucua | 17.08 | -97.61 | LakeSediment | MS | wNAm | Goman et al., 2018 |
| Lago Minucua | 17.08 | -97.61 | LakeSediment | varve | wNAm | Goman et al., 2018 |



| | | | | | | |
|---|---|---|---|---|---|---|
| Lago Puerto Arturo | 17.53 | -90.18 | LakeSediment | d18O | wNAm | Wahl et al., 2014 |
| Laguna De Aljojuca | 19.09 | -97.53 | LakeSediment | d18O | www.ncdc.noaa.gov/paleo/study/17735 | Bhattacharya et al., 2015 |
| Laguna de Juanacatlan | 20.63 | -104.74 | LakeSediment | Ti | wNAm | Jones et al., 2015 |
| Lake Elsinore | 33.67 | -117.35 | LakeSediment | d18O | www.ncdc.noaa.gov/paleo/study/30232 | Kirby et al., 2019 |
| Lake Elsinore | 33.67 | -117.35 | LakeSediment | particle size | www.ncdc.noaa.gov/paleo/study/30232 | Kirby et al., 2019 |
| Lake of the Woods | 43.48 | -109.89 | LakeSediment | stratigraphy | wNAm | Pribyl and Shuman, 2014 |
| Lake of the Woods | 49.05 | -120.18 | LakeSediment | chironomid | www.ncdc.noaa.gov/paleo/study/27330 | Palmer et al., 2002 |
| Lehman Caves | 39.00 | -114.22 | Speleothem | d13C | www.ncdc.noaa.gov/paleo/study/19038 | Steponaitis et al., 2015 |
| Lehman Caves | 39.00 | -114.22 | Speleothem | Mg/Ca | www.ncdc.noaa.gov/paleo/study/19038 | Steponaitis et al., 2015 |
| Leviathan | 37.89 | -115.58 | Speleothem | d13C | www.ncdc.noaa.gov/paleo/study/16517 | Lachniet et al., 2014 |
| Leviathan | 37.89 | -115.58 | Speleothem | d18O | www.ncdc.noaa.gov/paleo/study/16517 | Lachniet et al., 2014 |
| Lily | 59.20 | -135.40 | LakeSediment | pollen | www.ncdc.noaa.gov/paleo/study/15444 | Cwynar, 1990 |
| Lime Lake | 48.87 | -117.34 | LakeSediment | d18O | www.ncdc.noaa.gov/paleo/study/21250 | Steinman et al., 2016 |
| Lime Lake | 48.87 | -117.34 | LakeSediment | d13C | www.ncdc.noaa.gov/paleo/study/21250 | Steinman et al., 2016 |
| Little | 44.17 | -123.58 | LakeSediment | pollen | www.ncdc.noaa.gov/paleo/study/27330 | Worona and Whitlock, 1995 |
| Little Molas Lake | 37.74 | -107.71 | LakeSediment | pollen | wNAm | Toney and Anderson, 2006 |
| Little Windy | 41.43 | -106.33 | LakeSediment | stratigraphy | www.ncdc.noaa.gov/paleo/study/16096 | Minckley et al., 2012 |
| Logan | 60.58 | -140.50 | GlacierIce | d18O | www.ncdc.noaa.gov/paleo/study/15444 | Fisher et al., 2008 |
| Lone Fox Lake | 56.72 | -119.72 | LakeSediment | pollen | www.ncdc.noaa.gov/paleo/study/27330 | MacDonald and Cwynar, 1985 |
| Lonespruce | 60.01 | -159.14 | LakeSediment | BSi | www.ncdc.noaa.gov/paleo/study/15444 | Kaufman et al., 2012 |
| Louise Pond | 52.95 | -131.76 | LakeSediment | pollen | www.ncdc.noaa.gov/paleo/study/27330 | Pellatt and Mathewes, 1994 |
| Lowder Creek Bog | 37.66 | -112.77 | Peat | pollen | wNAm | Anderson RS et al., 1999 |
| Lower Bear Lake | 34.20 | -116.90 | LakeSediment | TOC | www.ncdc.noaa.gov/paleo/study/13215 | Kirby et al., 2012 |
| Lower Bear Lake | 34.20 | -116.90 | LakeSediment | C/N | www.ncdc.noaa.gov/paleo/study/13215 | Kirby et al., 2012 |
| M Lake | 68.27 | -133.47 | LakeSediment | pollen | www.ncdc.noaa.gov/paleo/study/27330 | Ritchie, 1977 |
| Macal Chasm | 16.88 | -89.11 | Speleothem | d13C | www.ncdc.noaa.gov/paleo/study/20506 | Akers et al., 2016 |
| Macal Chasm | 16.88 | -89.11 | Speleothem | d18O | www.ncdc.noaa.gov/paleo/study/20506 | Akers et al., 2016 |
| Macal Chasm | 16.88 | -89.11 | Speleothem | reflectance | www.ncdc.noaa.gov/paleo/study/20506 | Akers et al., 2016 |
| Marcella | 60.07 | -133.81 | LakeSediment | d18O | www.ncdc.noaa.gov/paleo/study/6066 | Anderson L. et al., 2007 |
| Marion | 49.31 | -122.55 | LakeSediment | pollen | wNAm | Mathewes, 1973 |
| MD02_2503 | 34.39 | -120.04 | MarineSediment | d18O | www.ncdc.noaa.gov/paleo/study/5582 | Hill et al., 2006 |





| | | | | | | |
|---|---|---|---|---|---|---|
| MD02_2515 | 27.48 | -112.07 | MarineSediment | alkenone | 10.1594/PANGAEA.861260 | McClymont et al., 2012 |
| MD02_2515 | 27.48 | -112.07 | MarineSediment | GDGT | 10.1594/PANGAEA.861260 | McClymont et al., 2012 |
| MD02-2499 | 41.65 | -124.94 | MarineSediment | diatom | www.ncdc.noaa.gov/paleo/study/24150 | Lopes and Mix, 2018 |
| Meli Lake | 68.68 | -149.08 | LakeSediment | d18O | www.ncdc.noaa.gov/paleo/study/5469 | Anderson L. et al., 2001 |
| Mexican Marin | 22.23 | -107.05 | MarineSediment | dD | www.ncdc.noaa.gov/paleo/study/24890 | Bhattacharya et al., 2018 |
| Mica Lake | 60.95 | -148.15 | LakeSediment | d18O | www.ncdc.noaa.gov/paleo/study/6202 | Schiff et al., 2009 |
| Midden Cluster 1 | 37.90 | -110.13 | Midden | macrofossils | geochange.er.usgs.gov/midden/ | Harbert and Nixon, 2018 |
| Midden Cluster 2 | 36.38 | -115.19 | Midden | macrofossils | geochange.er.usgs.gov/midden/ | Harbert and Nixon, 2018 |
| Midden Cluster 3 | 36.06 | -108.08 | Midden | macrofossils | geochange.er.usgs.gov/midden/ | Harbert and Nixon, 2018 |
| Midden Cluster 4 | 43.65 | -112.75 | Midden | macrofossils | geochange.er.usgs.gov/midden/ | Harbert and Nixon, 2018 |
| Midden Cluster 5 | 32.47 | -106.02 | Midden | macrofossils | geochange.er.usgs.gov/midden/ | Harbert and Nixon, 2018 |
| Midden Cluster 6 | 32.47 | -106.02 | Midden | macrofossils | geochange.er.usgs.gov/midden/ | Harbert and Nixon, 2018 |
| Midden Cluster 7 | 34.15 | -116.00 | Midden | macrofossils | geochange.er.usgs.gov/midden/ | Harbert and Nixon, 2018 |
| Midden Cluster 8 | 32.31 | -109.10 | Midden | macrofossils | geochange.er.usgs.gov/midden/ | Harbert and Nixon, 2018 |
| Midden Cluster 9 | 31.64 | -115.55 | Midden | macrofossils | geochange.er.usgs.gov/midden/ | Harbert and Nixon, 2018 |
| Minnetonka Cave | 42.09 | -111.52 | Speleothem | d13C | www.ncdc.noaa.gov/paleo/study/23097 | Lundeen et al., 2013 |
| Minnetonka Cave | 42.09 | -111.52 | Speleothem | d18O | www.ncdc.noaa.gov/paleo/study/23097 | Lundeen et al., 2013 |
| Moose Lake | 61.37 | -143.60 | LakeSediment | chironomid | www.ncdc.noaa.gov/paleo/study/15444 | Clegg et al., 2010 |
| Morris pond | 37.67 | -112.77 | LakeSediment | pollen | wNAm | Morris et al., 2013 |
| Mv0811-14JC | 34.30 | -120.00 | MarineSediment | stratigraphy | wNAm | Du et al., 2018 |
| MV99_PC14 | 25.20 | -112.72 | MarineSediment | Mg/Ca | www.ncdc.noaa.gov/paleo/study/10415 | Marchitto et al., 2010 |
| MV99-GC31 | 23.47 | -111.60 | MarineSediment | BSi | 10.1594/PANGAEA.824830 | Barron et al., 2012 |
| MV99-GC41/PC14 | 25.20 | -112.72 | MarineSediment | particle size | 10.1594/PANGAEA.896898 | Arellano-Torres et al., 2019 |
| Natural Bridge Caverns | 29.69 | -98.34 | Speleothem | Sr | wNAm | Wong et al., 2015 |
| Nevada Climate Division 3 | 37.80 | -115.80 | Wood | TRW | www.ncdc.noaa.gov/paleo/study/6384 | Hughes and Graumlich, 1996 |
| North Crater Lake | 49.07 | -120.02 | LakeSediment | chironomid | www.ncdc.noaa.gov/paleo/study/27330 | Palmer et al., 2002 |
| ODP_167_1019C | 41.68 | -124.93 | MarineSediment | alkenone | 10.1594/PANGAEA.841946 | Barron et al., 2003b |
| ODP1019 | 41.68 | -124.93 | MarineSediment | diatom | www.ncdc.noaa.gov/paleo/study/24150 | Lopes and Mix, 2018 |
| ODP1019 | 41.68 | -124.93 | MarineSediment | CaCO3 | www.ncdc.noaa.gov/paleo/study/5867 | Barron et al., 2003b |
| ODP1019 | 41.68 | -124.93 | MarineSediment | diatom | www.ncdc.noaa.gov/paleo/study/5867 | Barron et al., 2003b |


| ODP1019 | 41.68 | -124.93 | MarineSediment | pollen | www.ncdc.noaa.gov/paleo/study/5867 | Barron et al., 2003b |
| Oregon Caves | 42.08 | -123.42 | Speleothem | d13C | www.ncdc.noaa.gov/paleo/study/13543 | Ersek et al., 2012 |
| Oregon Caves | 42.08 | -123.42 | Speleothem | d18O | www.ncdc.noaa.gov/paleo/study/13543 | Ersek et al., 2012 |
| Oro Lake | 49.78 | -105.35 | LakeSediment | diatom | www.ncdc.noaa.gov/paleo/study/23073 | Michels et al., 2007 |
| OwensLake | 36.44 | -117.97 | LakeSediment | d18O | www.ncdc.noaa.gov/paleo/study/5472 | Benson et al., 2002 |
| P1B3 | 73.68 | -162.66 | MarineSediment | dinocyst | www.ncdc.noaa.gov/paleo/study/15444 | de Vernal et al., 2005 |
| Paradise | 54.69 | -122.62 | LakeSediment | d18O | www.ncdc.noaa.gov/paleo/study/21250 | Steinman et al., 2016 |
| Paradise | 54.69 | -122.62 | LakeSediment | d13C | www.ncdc.noaa.gov/paleo/study/21250 | Steinman et al., 2016 |
| Park Pond 1 | 43.47 | -109.96 | LakeSediment | pollen | www.ncdc.noaa.gov/paleo/study/27330 | Lynch, 1998 |
| Pink Panther | 32.08 | -105.17 | Speleothem | d18O | www.ncdc.noaa.gov/paleo/study/9739 | Asmerom et al., 2007 |
| Pixie | 48.60 | -124.20 | LakeSediment | pollen | www.ncdc.noaa.gov/paleo/study/27330 | Brown and Hebda, 2002 |
| Pixie Lake | 48.60 | -124.20 | LakeSediment | pollen | wNAm | Brown et al., 2006 |
| Posy | 37.94 | -111.70 | LakeSediment | pollen | www.ncdc.noaa.gov/paleo/study/27330 | Shafer, 1989 |
| Pyramid Lake | 40.07 | -119.58 | LakeSediment | d18O | www.ncdc.noaa.gov/paleo/study/5472 | Benson et al., 2002 |
| Quartz | 64.21 | -145.81 | LakeSediment | chironomid | www.ncdc.noaa.gov/paleo/study/15444 | Wooller et al., 2012 |
| Rainbow | 60.72 | -150.80 | LakeSediment | chironomid | www.ncdc.noaa.gov/paleo/study/15444 | Clegg et al., 2011 |
| RainbowLake | 44.94 | -109.50 | LakeSediment | stratigraphy | wNAm | Shuman and Marsicek, 2016 |
| Ranger | 67.15 | -153.65 | LakeSediment | pollen | www.ncdc.noaa.gov/paleo/study/15444 | Brubaker et al., 1983 |
| Rantin Lake | 60.03 | -129.03 | LakeSediment | CaCO3 | www.ncdc.noaa.gov/paleo/study/13095 | Pompeani et al., 2012 |
| Rapid | 42.73 | -109.19 | LakeSediment | pollen | www.ncdc.noaa.gov/paleo/study/27330 | Fall, 1988 |
| RC12-10 | 23.00 | -95.53 | MarineSediment | foraminifera | www.ncdc.noaa.gov/paleo/study/27330 | Poore et al., 2003 |
| Red Rock | 40.08 | -105.54 | LakeSediment | pollen | www.ncdc.noaa.gov/paleo/study/27330 | Maher, 1972 |
| Rhamnus Lake | 48.63 | -123.72 | LakeSediment | pollen | wNAm | Brown et al., 2006 |
| San Juan River Discharge | 48.58 | -124.31 | LakeSediment | pollen | wNAm | Brown and Schoups, 2015 |
| Schellings Bog | 40.28 | -123.36 | LakeSediment | pollen | wNAm | Barron et al., 2003a |
| Screaming Lynx Lake | 66.07 | -145.40 | LakeSediment | chironomid | www.ncdc.noaa.gov/paleo/study/15444 | Clegg et al., 2011 |
| Silver Lake | 35.37 | -116.14 | LakeSediment | particle size | www.ncdc.noaa.gov/paleo/study/20106 | Kirby et al., 2015 |
| Silver Lake | 35.37 | -116.14 | LakeSediment | C/N | www.ncdc.noaa.gov/paleo/study/20106 | Kirby et al., 2015 |
| Southern California | 33.77 | -116.66 | Peat | pollen | www.ncdc.noaa.gov/paleo/study/27330 | Ohlwein and Wahl, 2012 |
| Station 803 | 70.63 | -135.88 | MarineSediment | dinocyst | www.ncdc.noaa.gov/paleo/study/27910 | Bringué and Rochon, 2012 |
| Stella Lake | 39.01 | -114.32 | LakeSediment | chironomid | www.ncdc.noaa.gov/paleo/study/27330 | Reinemann et al., 2009 |





| Stewart Bog | 35.83 | -105.72 | Peat | pollen | wNAm | Jiménez-Moreno, et al., 2008 |
| Stowell Lake | 48.78 | -123.44 | LakeSediment | chironomid | www.ncdc.noaa.gov/paleo/study/27330 | Lemmen and Lacourse, 2018 |
| Swan Lake | 42.16 | -99.03 | LakeSediment | diatom | wNAm | Schmieder et al., 2011 |
| Takahula | 67.35 | -153.67 | LakeSediment | d18O | www.ncdc.noaa.gov/paleo/study/8663 | Clegg and Hu, 2010 |
| Tangled Up Lake | 67.67 | -149.08 | LakeSediment | d18O | www.ncdc.noaa.gov/paleo/study/5469 | Anderson L. et al., 2001 |
| Tiago Lake | 40.58 | -106.61 | LakeSediment | pollen | wNAm | Jiménez-Moreno, et al., 2011 |
| TN062-0550 | 40.87 | -124.57 | MarineSediment | pollen | www.ncdc.noaa.gov/paleo/study/27330 | Barron et al., 2018 |
| TN062-0550 | 40.87 | -124.57 | MarineSediment | diatom | www.ncdc.noaa.gov/paleo/study/27330 | Barron et al., 2018 |
| Trout Lake | 68.83 | -138.75 | LakeSediment | chironomid | www.ncdc.noaa.gov/paleo/study/15444 | Irvine et al., 2012 |
| Upper Big Creek Lake | 40.91 | -106.62 | LakeSediment | stratigraphy | wNAm | Shuman et al., 2015 |
| Upper Fly | 61.07 | -138.09 | LakeSediment | pollen | www.ncdc.noaa.gov/paleo/study/15444 | Bunbury and Gajewski, 2009 |
| Upper Pinto Fen | 53.58 | -118.02 | Peat | DBD | www.ncdc.noaa.gov/paleo/study/13665 | Yu et al., 2003 |
| W8709-13PC | 42.12 | -125.75 | MarineSediment | diatom | www.ncdc.noaa.gov/paleo/study/24150 | Lopes and Mix, 2018 |
| WA01 | 61.24 | -136.93 | LakeSediment | TOC | www.ncdc.noaa.gov/paleo/study/18435 | Rainville and Gajewski, 2013 |
| Waskey Lake | 59.88 | -159.21 | LakeSediment | TOC | www.ncdc.noaa.gov/paleo/study/15444 | Levy et al., 2004 |
| Windy Lake | 49.81 | -117.88 | LakeSediment | chironomid | www.ncdc.noaa.gov/paleo/study/27330 | Chase et al., 2008 |
| Wolverine Lake | 67.10 | -158.91 | LakeSediment | MAR | www.ncdc.noaa.gov/paleo/study/23070 | Mann et al., 2002 |
| Yellow Lake | 39.65 | -107.35 | LakeSediment | d18O | www.ncdc.noaa.gov/paleo/study/13120 | Anderson L., 2012 |

[a]Abbreviations for proxy types: biogenic silica (BSi), calcium carbonate ($CaCO_3$), dry bulk density (DBD), glycerol dialkyle glycerol tetraethers (GDGT), mass accumulation rate (MAR), magnesium/calcium (Mg/Ca), sulfur (S), strontium (Sr), total organic carbon (TOC), tree-ring width (TRW), titanium (Ti), carbon 13 isotopes ($\delta^{13}C$), oxygen 18 isotopes ($\delta^{18}O$), and deuterium isotopes of leaf wax (dD).

**Supplementary Table 1:** Essential metadata for records in the western North America Holocene paleoclimate database, with links to data in LiPDverse. See text for explanation of fields.

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
