# Peer review of "A multiproxy database of western North American"

_Earth System Science Data, 2020_

## Referee Comment (RC1) · Anonymous Referee #1 · 20 Oct 2020

This paper presents an impressive compilation of previously published western North American paleoclimate records (marine and continental) covering the whole Holocene. This database is well-structured and compiled in Linked Paleo Data (LiPD) format that can be read with Matlab, Pyton and R (I used the R_version). In addition, data can be also visualized and downloaded through the website "LiPDverse". This is very helpful for people that are not used to MatLab, Python, or R languages. I acknowledge the metadata compilation that helps the data re-use. I totally recommend its publication and I only have some minor comments.

My first comment is related to the chronology of the sites. I'm aware that this compilation provides the original radiocarbon dates (as well as the age reservoir used in marine sediments) so that age models can be updated with new radiocarbon calibra-

tions. However, when one visualizes or downloads the data from each individual site (e.g, by means of the LiPDverse tools) there should be a reference to the calibration curve used for that specific age model.

I detected a potential problem in the LiPD Utilities_R installation that I do not know if it is related to the R version I used (the latest R release: R version 4.0.3 2020-10-10 working on MacOS Catalina). After installing devtools [install.packages("devtools")], and before loading devtools package [library(devtools)] I had to install the package usethis [install.packages("usethis")] and loaded this package [library(usethis)], otherwise, I always got the error "Loading required package: usethis" that did not allow me to install the LiPD Utilities.

Line 235-236. Please correct the link: add one slash between wNAm and 0_15_0: http://lipdverse.org/wNAm/0_15_0/

---

## Referee Comment (RC2) · Jessie Woodbridge (Referee) · 2 Nov 2020

This paper presents a thorough and valuable amalgamation of palaeoclimate datasets, which have been transformed into a usable format. The detailed metadata included will be highly beneficial for any users of the data and code. This work represents a great example of open science with benefit to the wider scientific community. Specific comments below: Line 64: Are there other reasons that could be mentioned for studying and amalgamating records of Holocene climate? For example, a period with increasing human impacts when modern climate patterns were established and modern coats lines formed, etc... Line 94: 2.1 Data collection - "were considered" - This sentence would benefit from addition of "for inclusion in the database". Line 114: How were the data 'digitized'? Line 125: How much do site-specific characteristics influence proxy

relationships with climate? Line 150: Was a minimum number of dates used as criteria for inclusion of records? Line 187: "interpreted in a peer-reviewed publication" – what publication is this and where is this available? Line 239: Were the climate reconstructions performed by the authors or the original analysts? Line 317: Why was 500 year binning selected? How and why were the grid sizes selected? Does changing these factors alter patterns and interpretations? Maybe state here whether age-depth reconstructions are available in the database and how these were developed. Line 329: The statement "help identify future research priorities" would benefit from elaboration. Line 346: When I follow this link: http://lipdverse.org/wNAm0_15_0 I receive a message saying "page not found". Fig 3: How much confidence is there in climate reconstructions when there are very few records for certain time periods? What is the impact on the overall patterns observed of sites appearing/disappearing through the time series depending on the time frame specific records cover? Fig 4: Could numbers and labels be added to the left side y-axis on the lower graph as well?

There is potential for inclusion of extra text and references that could be included in relation to reviewing existing literature and wider discussion around the specific datasets and climatic interpretations (if this is in line with this style of paper). It would also be useful to mention that pollen datasets can be influenced by human land use. While such impacts will be minor in North America for the majority of the Holocene, the impact of people upon vegetation still deserves some acknowledgement, as this can complicate the climate signal in the latter Holocene. For example, see the work of the ArchaeoGlobe group: https://science.sciencemag.org/content/365/6456/897

---

## Author Comment (AC1) · 23 Dec 2020

Thank you for your positive review of our manuscript. In response to including calibration-curve information for each record: we are focusing our synthesis on what's most useful for reusability, not reiterating all the information contained in the underlying studies. The calibration curve is one of the many choices and variables that go into generating an age model. More important than the calibration curve is the choice of the age-model and the parameters behind it. Even more important is which ages are retained and what material was dated. We chose to invest our time in collecting the primary underlying data for reuse, noting that for most applications, being able to recalibrate or update calibrations is more practical than filtering or aligning records based on original calibrations.

[Figure]

In response to the problems encountered installing the R LiPD Utilities, we appreciate your enthusiasm for vetting the R-package installation. The installation problem you encountered has now been resolved. We have also tested the installation on R version 4.0.3 2020-10-10 and MacOS Catalina. We will update the website installation instructions accordingly. In brief, entering the following commands in R Studio will install the R LiPD utility package.

install.packages("remotes")   remotes::install_github("nickmckay/lipd-utilities",subdir = "R") library(lipdR)

Finally, we have also corrected the lipdverse link: http://lipdverse.org/wNAm/0_15_0

---

## Author Comment (AC2) · 23 Dec 2020

Thank you for your thorough review of our manuscript. We will address the manuscript flow and clarity points raised as appropriate. These include the specific wording suggestions as well as minor points of clarification as follows.

Line 64: Yes, we will include this example

Line 94: We will modify the sentence as suggested

Line 114: Data were digitized using the MATLAB program digitize2.m from the MATLAB file exchange website: https://www.mathworks.com/matlabcentral/fileexchange/928-digitize2-m. This will be clarified in the manuscript.

[Figure]

Line 125: It is often unclear the extent to which site-specific characteristics can influence proxy relationships with climate. This information is sometimes discussed in the original publications; however, we think that gathering the evidence and evaluating patterns across multiple sites is a useful way to reduce site-specific uncertainties.

Line 150: As stated, the number of dates required are five relatively evenly distributed Holocene dates. Alternatively, a minimum of 3000 years between chronology tie-points is required.

Line 187: "interpreted in a peer-revied publication" refers to the original publications that produced the proxy records, not a previous synthesis product. This point belongs, and is already discussed, in section 2.2. For clarification, we will delete this sentence and focus on metadata.

Line 239: The climate reconstructions were performed by the original analysts except in the case of the midden reconstructions. In these cases, precipitation reconstructions were performed by us on subsets of the midden clusters analyzed by the original authors (Harbert et al., 2018). We used the same midden cluster subsets as the temperature reconstructions developed in Kaufman et al. (2020). We will add text to clarify this and elaborate on the specific method applied.

Line 317: The 500-year bin-size was selected somewhat arbitrarily to help showcase Holocene trends in database. Finer bins show the same long-term patterns with increasing high-frequency variability, because fewer and fewer records contribute to each bin. Grid sizes were selected similarly, to help account for the uneven spatial distribution of records, while following previously published examples.

Line 329: We will elaborate here. Future research priorities for example include focusing new record development in data-sparse regions.

Line 346: We will fix the lipdverse link, which was missing a forward slash between wNAm and 0_15_0. http://lipdverse.org/wNAm/0_15_0. This link will be updated to

version 1.0.0 upon the official publication of the database.

Fig 3: These are not reconstructions, rather composites by proxy-type to illustrate the database contents. Confidence in these patterns is low where there are few contributing records. This is reflected in part by the wider 95% error bars on estimates of the mean.

Fig 4: We will move the axes numbering to the left side of the lower panels.

In response to adding extra text and references around specific datasets and interpretations: Our intention with this manuscript is to gather the available evidence. It is beyond the scope of this effort to adequately evaluate the strengths and weaknesses of specific datasets. Similarly, strengths and weaknesses of specific proxy types are equally extensive. Human influence on Holocene pollen datasets is one of many factors influencing specific proxy types. To direct readers to learn more, we will add the text: "Background information including the strengths, weaknesses, and underlying assumptions of the specific poxy types can be found in textbooks devoted to the topic (e.g., Bradley, 2015)". Nonetheless, we agree that future analysis publications toward evaluating specific dataset relationships with climate and the strengths and weaknesses of different proxy types are warranted.

References:

Bradley, R. S. Paleoclimatology: reconstructing climates of the Quaternary. Elsevier, 2015.

Harbert, R. S. and Nixon, K. C.: Quantitative Late Quaternary climate reconstruction from plant macrofossil communities in western North America, Open Quaternary, 4(1), 8, doi:10.5334/oq.46, 2018.

Kaufman, D., McKay, N., Routson, C., Erb, M., Davis, B., Heiri, O., Jaccard, S., Tierney, J., Dätwyler, C., Axford, Y., Brussel, T., Cartapanis, O., Chase, B., Dawson, A., de Vernal, A., Engels, S., Jonkers, L., Marsicek, J., Moffa-Sánchez, P., Morrill, C., Orsi, A.,

[Figure]

Rehfeld, K., Saunders, K., Sommer, P. S., Thomas, E., Tonello, M., Tóth, M., Vachula, R., Andreev, A., Bertrand, S., Biskaborn, B., Bringué, M., Brooks, S., Caniupán, M., Chevalier, M., Cwynar, L., Emile-Geay, J., Fegyveresi, J., Feurdean, A., Finsinger, W., Fortin, M.-C., Foster, L., Fox, M., Gajewski, K., Grosjean, M., Hausmann, S., Heinrichs, M., Holmes, N., Ilyashuk, B., Ilyashuk, E., Juggins, S., Khider, D., Koinig, K., Langdon, P., Larocque-Tobler, I., Li, J., Lotter, A., Luoto, T., Mackay, A., Magyari, E., Malevich, S., Mark, B., Massaferro, J., Montade, V., Nazarova, L., Novenko, E., Pařil, P., Pearson, E., Peros, M., Pienitz, R., Płóciennik, M., Porinchu, D., Potito, A., Rees, A., Reinemann, S., Roberts, S., Rolland, N., Salonen, S., Self, A., Seppä, H., Shala, S., St-Jacques, J.-M., Stenni, B., Syrykh, L., Tarrats, P., Taylor, K., van den Bos, V., Velle, G., Wahl, E., Walker, I., Wilmshurst, J., Zhang, E. and Zhilich, S.: A global database of Holocene paleotemperature records, Sci. Data, 7(1), 115, doi:10.1038/s41597-020-0445-3, 2020.